# Vapor-Phase Oxidant-Free Dehydrogenation of 2,3- and 1,4-Butanediol over Cu/SiO$_2$ Catalyst Prepared by Crown-Ether-Assisted Impregnation

Enggah Kurniawan, Shuya Hosaka, Masayuki Kobata, Yasuhiro Yamada and Satoshi Sato *

Graduate School of Engineering, Chiba University, Yayoi, Inage, Chiba 263-8522, Japan
* Correspondence: satoshi@faculty.chiba-u.jp; Tel.: +81-43-290-3377

**Abstract:** A silica-supported copper (Cu/SiO$_2$) catalyst containing highly dispersed Cu nanoparticles was prepared via a crown-ether-assisted impregnation method. A 12-crown-4-ether-assisted Cu/SiO$_2$ catalyst outperformed several Cu/SiO$_2$ catalysts prepared with various organic additives in the dehydrogenation of 2,3- and 1,4-butanediol. It was found that the catalytic activity, i.e., the formation rate of acetoin from 2,3-butanediol and that of γ-butyrolactone from 1,4-butanediol, was proportional to the copper surface area.

**Keywords:** organic additive; crown ether; Cu/SiO$_2$ catalyst; butanediol dehydrogenation

## 1. Introduction

With the growing demand for cosmetics, foods, and chemical products, the need for platform chemicals that become the backbone of these industries is rapidly increasing [1–3]. Since most platform chemicals are supplied from petrol-based compounds originating from depleting fossil fuels, the growing demand for platform chemicals increases fossil fuel consumption, potentially impacting the environment [4]. Shifting the origin of platform chemicals from fossil fuels to greener and more abundant resources such as biomass is considered a viable solution [5–7]. Due to this reason, the catalytic technology to transform biomass-derived compounds into value-added chemicals is economically and industrially valuable.

Butanediols, such as 2,3- and 1,4-butanediol (2,3- and 1,4-BDO), have received considerable interest due to the emerging sustainability of their bio-based production through the metabolic conversion of biomass [8–14]. In addition, 2,3- and 1,4-BDO derivatives have been used in the food, cosmetics, plasticizers, apparel, and polymer industries [8,12,15]. An efficient catalytic design that utilizes the sustainable nature of 2,3- and 1,4-BDO to obtain their derivatives is surely in line with the effort to alleviate fossil fuel consumption.

Catalytic oxidant-free dehydrogenation is a powerful strategy to obtain value-added BDO derivatives such as acetoin (3-hydroxy-2-butanone, AC) and γ-butyrolactone (GBL). AC is essential for flavor enhancers, fragrances, and chelating agents [16]. Meanwhile, GBL is a main precursor of pyrolidones, herbicides, and rubber additives [17,18]. Catalytic oxidant-free dehydrogenation is also greener than the conventional oxidation protocol using stoichiometric Cr or Mn salts [19–23]. Furthermore, many reports have suggested that oxidant-free dehydrogenation of biomass-derived alcohols is a green and atom-efficient strategy for hydrogen production [19,24–29]. Due to the above-mentioned reasons, a highly efficient catalyst for the oxidant-free dehydrogenation of BDO to AC and GBL is highly desired.

Extensive work has been dedicated to designing highly efficient catalysts for liquid-phase dehydrogenation of alcohol to aldehyde or ketone using precious and non-precious metals [30–32]. Inexpensive and abundant Cu metal emerges as efficient catalysts in numerous catalytic dehydrogenations of alcohols [33–35]. Therefore, Cu metal could be

potentially employed as an efficient catalyst for oxidant-free dehydrogenation of BDO. As an alternative to liquid-phase catalysis, vapor-phase catalytic systems have been reported to offer some advantages, such as simple operation, separation, and catalyst recycling [36]. Combining these advantages with an efficient Cu catalyst for the dehydrogenation of BDO is highly beneficial from an industrial and environmental point of view.

Cu-based catalysts have been employed for the dehydrogenation of BDOs. Recently, supported Cu catalysts, such as $Cu/SiO_2$ [37,38], $Cu-Al_2O_3$ [39], and Li-CuZnAl [40], have been used as efficient catalysts for the dehydrogenation of 2,3-BDO to AC (Scheme 1a). Nevertheless, several issues, such as a high reaction temperature, low productivity, and long contact time, still need to be addressed. Numerous reports have also shown that Cu-based catalysts are active in the dehydrogenation of 1,4-BDO to GBL (Scheme 1b) [17,41–45]. However, the reported Cu catalysts still suffer from low GBL productivity. Several research groups have studied Cu catalysts for GBL formation via a coupling reaction of 1,4-BDO dehydrogenation with the hydrogenation of maleic anhydride [46], furfural [47], nitrobenzene [48], and benzaldehyde [49]. In addition, a Cu-based catalyst, namely the Cu/Zn/Zr/Al catalyst, is also active for the dehydrogenation of 1,2- and 1,3-BDO [50,51]. In the dehydrogenation of 1,2-BDO, 1-hydroxy-2-butanone is produced selectively, although high selectivity to 4-hydroxy-2-butanone from 1,3-BDO cannot be obtained due to the formation of several by-products, such as butanone and acetone. Employing a less acidic Cu-based catalyst might provide better selectivity in the dehydrogenation of 1,3-BDO to 4-hydroxy-2-butanone. According to the above reports, several issues relating to the efficiency of the catalyst can still be addressed despite the promising activity of Cu-based catalysts in the dehydrogenation of BDOs.

**Scheme 1.** Dehydrogenation of 2,3-BDO to AC (**a**) and 1,4-BDO to GBL (**b**).

Many reports have suggested that the efficiency of Cu catalysts is strongly correlated with their Cu dispersion, surface area, and particle size [52–54]. Numerous strategies have been developed to prepare Cu catalysts that contain highly dispersed Cu nanoparticles [52–55]. In line with this effort, we have also reported a strategy to generate highly active Ni-based catalysts containing highly dispersed Ni nanoparticles using an organic-additive-assisted impregnation protocol [56–58]. This strategy has been proven to improve the catalytic activity of Ni catalysts for the hydrogenation of $CO_2$ [56], levulinic acid [57], and acetoin [58]. Recently, we have also prepared $Cu/SiO_2$ catalysts using various organic additives for the vapor-phase dehydrogenation of 1-decanol to decanal [59]. The stability of the complex formed between crown ether and $Cu^{2+}$ during catalyst preparation influenced the Cu dispersion and particle size, consequently affecting the catalyst efficiency [59]. A crown-ether-assisted $Cu/SiO_2$ efficiently catalyzed the dehydrogenation of 1-decanol, even though 1-decanol as a primary aliphatic alcohol has been reported to be

relatively stable [22,60]. Therefore, efficient organic-additive-assisted $Cu/SiO_2$ catalysts might show promising activity for the oxidant-free dehydrogenation of 2,3- and 1,4-BDO.

In this report, $Cu/SiO_2$ catalysts prepared via a crown-ether-assisted impregnation method were employed in the vapor-phase oxidant-free dehydrogenation of 2,3- and 1,4-BDO. The catalytic activity of several $Cu/SiO_2$ catalysts prepared with various organic additives such as 12-crown-4-ether (12C4), 15-crown-5-ether (15C5), 18-crown-6-ether (18C6), triethylene glycol (TEG), and citric acid (CA) was examined. The preparation using organic additives substantially improved the Cu surface area, thus significantly enhancing catalytic activity of the respective Cu catalyst. The formation rates of AC and GBL are proportional to the surface area of Cu. This simple and effective strategy can be viewed as an alternative to upgrading biomass-derived platform compounds to value-added chemicals.

## 2. Materials and Methods

### 2.1. Catalyst Preparation

Crown ethers such as 12C4, 15C5, and 18C6 were purchased from Tokyo Chemical Industries Co., Ltd., Tokyo, Japan. TEG, CA, and $Cu(NO_3)_2 \cdot 3H_2O$ were purchased from Wako Pure Chemical Industries, Ltd., Osaka, Japan. Silica-supported copper ($Cu/SiO_2$) catalysts were prepared by an impregnation method using an aqueous solution containing a prescribed amount of $Cu(NO_3)_2 \cdot 3H_2O$ and an organic additive; the molar ratio of the organic additive to the metal is 1. Typically, $Cu(NO_3)_2 \cdot 3H_2O$ (0.152 g, 0.63 mmol) and 12C4 (0.110 g, 0.63 mmol) were dissolved in 20 mL of $H_2O$. The solution was dropped onto 1.96 g of $SiO_2$ support, which was supplied by Fuji Silysia Chemical Ltd., Kasugai, Japan (CARiACT Q6 with a granule size of 75–500 μm and a specific surface area ($SA$) of 451 $m^2 \ g^{-1}$), and the water of the dropped solution was evaporated by illuminating the support with a 350-W electric light bulb at the support surface temperature of ca. 70 °C. This process was repeated until all the copper nitrate solution had been added. The resultant sample was dried at 110 °C for 12 h and calcined at 300 °C for 3 h to obtain a $CuO/SiO_2$ sample. Prior to the catalytic reaction and characterization, the resultant $CuO/SiO_2$ sample was reduced in $H_2$ flow at 300 °C for 1 h to prepare $Cu/SiO_2$. The resulting catalyst is expressed as $A$-$x$$Cu/SiO_2$, where $A$ and $x$ indicate the used organic additive and the weight percentage of Cu metal, respectively. In this case, the above procedure generated a 12C4-$2Cu/SiO_2$ catalyst. The $x$$Cu/SiO_2$ catalysts with different Cu contents were prepared by changing the quantities of Cu salt, organic additive, and $SiO_2$ support, which are described in Table S1 of the supporting information (SI) file.

### 2.2. Catalyst Characterization

The thermal decomposition profile with mass analysis of thermally evolved gases (TD-MS) was recorded on the BELLMASS unit (Microtrac BEL Corp., Osaka, Japan) to identify the components of the produced gasses. A sample (50 mg) was heated from 100 to 800 °C in a He flow of 30 $cm^3 \ min^{-1}$ at 10 $°C \ min^{-1}$ after the sample had been heated in the He flow at 110 °C for 1 h and then cooled to 100 °C, where the preheating process was performed to remove pre-adsorbed gasses. The produced gases were monitored using a thermal conductivity detector (TCD) and a quadrupole mass analyzer. The thermogravimetry (TG) analysis was conducted using the Thermoplus 8120E2 (Rigaku Corp., Tokyo, Japan). The sample of ca. 7 mg was heated from room temperature to 110 °C, and the temperature of 110 °C was maintained for 1 h to remove surface water adsorbed on the catalyst. After that, the weight change of the sample was recorded while the sample was heated from 110 to 900 °C in the air at a heating rate of 5 $°C \ min^{-1}$. The X-ray diffraction measurement (XRD) was performed on a Miniflex 600 (Rigaku Corp.) using monochromatic Cu Kα (λ = 0.15418), equipped with a D/tex Ultra 250 ID detector. The Cu surface area, $SA_{Cu}$, was estimated by TPR titration of oxidized Cu by $N_2O$ at 50 °C according to the previously reported method [61]. The details of this method are described in the SI file.

## 2.3. Vapor-Phase Dehydrogenation of 2,3-BDO and 1,4-BDO

Vapor-phase dehydrogenation of 2,3-BDO was performed in a fixed-bed flow reactor at an atmospheric pressure of $N_2$. The catalyst was loaded into the reactor and reduced with $H_2$ gas at 300 °C for 1 h. Thereafter, 2,3-BDO was fed through the reactor top at a liquid feed rate of 1.37 g $h^{-1}$ together with an $N_2$ flow of 30 $cm^3$ $min^{-1}$ at 200 °C for the time on stream (TOS) of 5 h. The reaction effluents were collected in a dry ice-acetone trap every hour. The recovered products were identified using a gas chromatograph (GC) equipped with a mass-spectrometer (QP5050A, Shimadzu Corp., Kyoto, Japan) and a capillary column (InertCap-WAX, a length of 30 m with an inner diameter (ID) of 0.25 mm, GL-Science Inc., Tokyo, Japan). They were quantitatively analyzed by a GC (GC-8A, Shimadzu Corp.) equipped with a flame ionization detector and a 30-m capillary column of Inert Cap 1 (ID of 0.53 mm and a length of 30 m, GL-Science Inc.) using 1-hexanol as an internal standard.

The vapor-phase dehydrogenation of 1,4-BDO was also performed in a similar way to the 2,3-BDO dehydrogenation under the following conditions: reduction in $H_2$ flow at 300 °C for 1 h, 1,4-BDO feed at a liquid feed rate of 1.80 g $h^{-1}$ together with an $N_2$ flow of 30 $cm^3$ $min^{-1}$. The reaction effluents were collected in an ice-water trap every hour. The recovered products were quantitatively analyzed by a GC (GC-2014, Shimadzu Corp.) equipped with an FID and a 30-m capillary column of InertCap 1 (ID of 0.25 mm, GL-Science Inc.) using 1-hexanol as an internal standard.

## 3. Results

### 3.1. Catalyst Characterization

Figure 1a,b show the TD-MS profiles of none- and 12C4-2Cu/$SiO_2$ catalysts. The $m/z$ signals of 18, 30, 14, and 28 were associated with $H_2O$, NO, N, and CO, respectively. The signals at $m/z$ = 43 in the as-prepared 12C4-2Cu/$SiO_2$ would be attributed to $CH_2CHO$, and the $m/z$ of 44 originated from $CH_2CH_2O$ and $CO_2$. On the other hand, the signal at $m/z$ = 44 in the as-prepared none-2Cu/$SiO_2$ might originate from $CO_2$. It can also be seen that the decomposition of nitrate was primarily observed in the as-prepared none-2Cu/$SiO_2$ sample. Meanwhile, the signals corresponding to $CH_2CHO$ and $CH_2CH_2O$+$CO_2$ in the 12C4-2Cu/$SiO_2$ sample were stronger than those in the none-2Cu/$SiO_2$ sample. This finding suggests the presence of a larger organic molecule attached to the Cu cation in the as-prepared 12C4-2Cu/$SiO_2$ sample. The TG curves of the as-prepared none- and 12C4-2Cu/$SiO_2$ samples are depicted in Figure 1c. Notably, the weight loss for the 12C4-2Cu/$SiO_2$ sample was larger than that for the none-2Cu/$SiO_2$ sample, which is in good agreement with the results from TD-MS.

Figure 2a shows the XRD patterns of 2Cu/$SiO_2$ catalysts prepared with different organic additives. Small peaks associated with the monoclinic CuO phase were observed at $2\theta$ of 35.6°, 38.7°, and 48.8° in the 2Cu/$SiO_2$ catalyst prepared without organic additive [62,63]. The XRD profiles of the freshly reduced none- and 12C4-2Cu/$SiO_2$ catalysts are shown in Figure 2b. According to the XRD profiles, the reduction protocol has transformed the CuO species into metallic Cu, as the peaks corresponding to CuO were no longer observed and the peaks associated with metallic Cu appeared at $2\theta$ of 43.3° (111) and 50.4° (200) [64]. It is worth noting that the freshly reduced 12C4-2Cu/$SiO_2$ catalyst has a weaker peak intensity than the none-2Cu/$SiO_2$. Previous reports suggest that this phenomenon is probably due to the presence of highly dispersed Cu nanoparticles [53,54]. In addition, the $N_2O$ titration clearly revealed that the Cu dispersions, $D$, of the 12C4- and none-2Cu/$SiO_2$ catalysts were 0.319 and 0.124, and that the $SA_{Cu}$ values of the 12C4- and none-2Cu/$SiO_2$ catalysts were 3.53 and 1.38 $m^2$ $g^{-1}$, respectively, which are comparable to the reported value (1.9 $m^2$ $g^{-1}$) of 1.4 wt.-% Cu/$SiO_2$ [64]. Therefore, the XRD and $N_2O$ titration suggest that the utilization of 12C4 as an organic additive generated the Cu/$SiO_2$ with a higher Cu dispersion and possibly smaller Cu size.

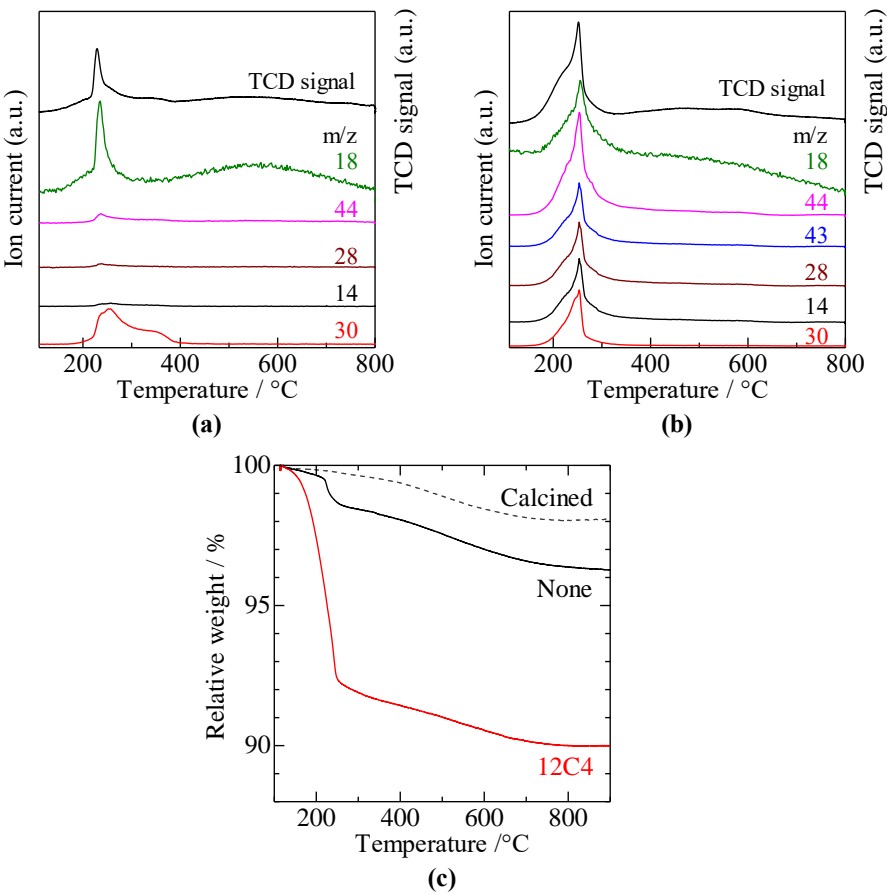

**Figure 1.** TD-MS profiles of the as-prepared none- (**a**) and 12C4-2Cu/SiO₂ (**b**) and TG curves of the as-prepared none-, as-prepared 12C4-, and calcined none-2Cu/SiO₂ (**c**).

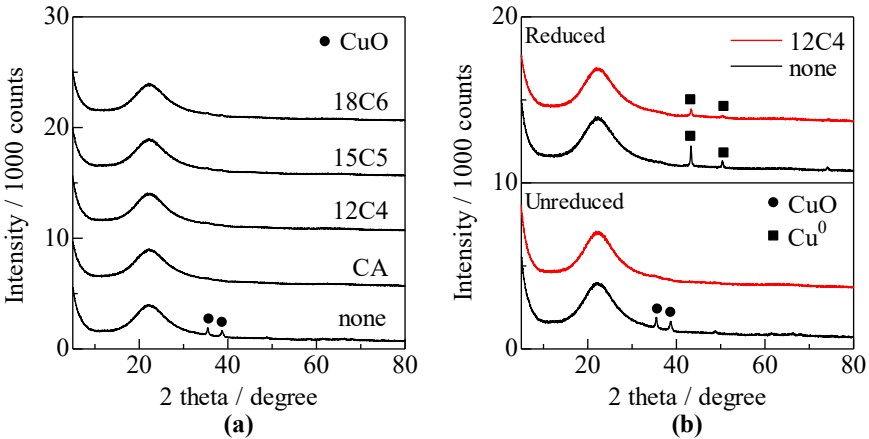

**Figure 2.** XRD profiles of 2Cu/SiO₂ catalysts prepared with different organic additives before reduction (**a**) and none- and 12C4-2Cu/SiO₂ catalysts (**b**).

*3.2. Comparison of Organic Additives*

Table 1 shows the catalytic activity of several 2Cu/SiO₂ catalysts in the dehydrogenation of 2,3-BDO to AC at 200 °C and a contact time, $W/F$, of 0.18 h, where $W/F$ is defined as the weight of the catalyst, $W$, divided by the feed rate of the reactant, $F$, which represents the time required for the weight turnover to reach 1 [65]. All Cu catalysts exhibit outstanding selectivity to AC (>98%), with only DA as the side product. A small 2,3-BDO conversion of 19% was observed when the dehydrogenation was catalyzed by a none-2Cu/SiO₂ catalyst (entry 1). Among several organic additives tested, 12C4 generated

Cu/SiO₂ catalysts with the highest catalytic performance (entry 2). Other crown ethers, such as 15C5 and 18C6, also generated 2Cu/SiO₂ catalysts with higher performance than a none-2Cu/SiO₂ catalyst (entries 3 and 4), even though those catalysts were inferior to the 12C4-2Cu/SiO₂ catalyst. CA, which was efficient in improving the activity of Ni-based catalysts [56–58], also improved the catalytic performance of the 2Cu/SiO₂ catalyst (entry 5). Nevertheless, the performance of the CA-2Cu/SiO₂ catalyst was slightly lower than that of the 12C4-2Cu/SiO₂ catalyst.

**Table 1.** Effect of organic additive on the catalytic performance of 2Cu/SiO₂ catalyst in the dehydrogenation of 2,3-BDO.

| Entry | Organic Additive | $D$ | $SA_{Cu}$/ $m^2\ g^{-1}$ | Conversion [a]/ mol% | Selectivity [a]/mol% | |
|---|---|---|---|---|---|---|
| | | | | | AC | DA |
| 1 | none | 0.248 | 2.76 | 19.0 | 99.3 | 0.0 |
| 2 | 12C4 | 0.632 | 7.06 | 56.4 | 98.8 | 1.0 |
| 3 | 15C5 | 0.613 | 6.84 | 43.0 | 99.2 | 0.5 |
| 4 | 18C6 | 0.521 | 5.81 | 38.9 | 99.3 | 0.4 |
| 5 | CA | 0.549 | 7.12 | 50.2 | 99.0 | 1.0 |

Reaction conditions: Reaction temperature, 200 °C; $W/F$, 0.18 h. [a] Average conversion and selectivity at TOS of 0–1 h.

The XRD and $SA_{Cu}$ measurements indicate that the utilization of organic additives generated 2Cu/SiO₂ catalysts with better Cu dispersion and smaller Cu sizes, leading to a higher Cu surface area. The improvement of the catalytic activity of the 2Cu/SiO₂ catalysts was strongly correlated with the $D$ and $SA_{Cu}$. The 12C4-2Cu/SiO₂ catalyst possessed the highest $D$ and $SA_{Cu}$; therefore, this catalyst gave the highest catalytic performance among several 2Cu/SiO₂ catalysts tested. It has been reported that the efficiency of the crown-ether-assisted Cu/SiO₂ catalysts was influenced by the stability of the complex formed between Cu²⁺ and crown ether: Table S2 cites data for the $SA_{Cu}$ of several Cu/SiO₂ catalysts prepared with and without organic additives [59]. Cu²⁺ was reported to form a more stable complex with 12C4 than with 15C5 and 18C6 [66,67]; consequently, 12C4 was able to prevent Cu agglomeration during the calcination process more effectively than 15C5 and 18C6. The trend of catalytic performance of 12C4- and CA-2Cu/SiO₂ catalysts was also in good agreement with their corresponding $SA_{Cu}$.

*3.3. Effect of Reaction Conditions such as W/F, Temperature, and N₂ Flow Rate*

Figure 3a shows the effect of $W/F$ on the catalytic activity of the 12C4-2Cu/SiO₂ catalyst at a reaction temperature of 200 °C. Prolonging the $W/F$ increased the activity of the 12C4-2Cu/SiO₂ catalyst. The AC selectivity remained above 98% even at the longest $W/F$ tested. This result suggests that the consecutive dehydrogenation of AC to DA was not affected by the $W/F$. The highest yield of AC generated at a $W/F$ of 1.09 h and 200 °C was 79.2%, and further increments of $W/F$ gave a negligible increase in the AC yield.

Figure 3b illustrates the effect of temperatures on the equilibrium yield in the dehydrogenation of 2,3-BDO in a N₂ flow of 30 cm³ min⁻¹. The increase in temperature increased the conversion of 2,3-BDO but decreased the AC selectivity. The AC selectivities were 99.2 and 97.6% at 170 and 200 °C, respectively. Meanwhile, the reaction temperatures of 250 and 275 °C gave 87.7 and 75.8% AC selectivity, respectively. The decline in AC selectivity was due to the consecutive dehydrogenation of AC to DA, as the DA selectivity increased with the increase in temperature. The dehydrogenation of 2,3-BDO to AC is an equilibrium reaction (Scheme 1a); therefore, the pressure equilibrium constant of the 2,3-BDO dehydrogenation at a given reaction temperature can be expressed as:

$$K_{1p} = \frac{P_{AC} \times P_{H_2}}{P_{2,3-BDO}}$$

where $P_i$ is the partial pressure of component i. The $K_{1p}$ values were estimated to be 0.138, 0.422, 1.83, and 2.26 in the dehydrogenation-hydrogenation reaction between 2,3-BDO and AC at 170, 200, 250, and 275 °C, respectively (the broken red line in Figure 3b). It is worth noting that the $K_{1p}$ values for the reaction temperature of 170–250 °C are similar to the inverse of the pressure equilibrium constant in the AC hydrogenation ($1/K_{3p}$) [59] (black dotted line). However, the $K_{1p}$ value at 275 °C deviated from that of the $1/K_{3p}$ due to a high rate of AC dehydrogenation to DA, as confirmed later by Figure 3d.

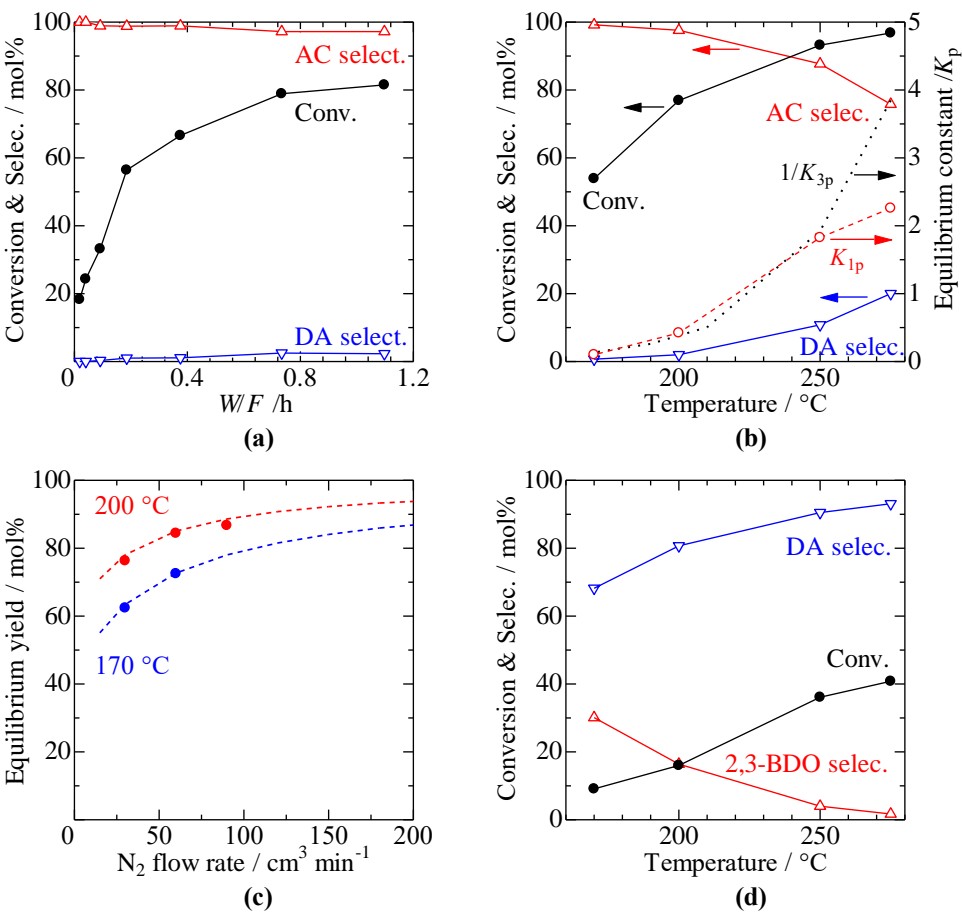

**Figure 3.** Effects of contact time, $W/F$, (**a**), reaction temperature at $W/F$ = 1.5 (170 °C), 0.76 (200 °C), 0.36 (250 °C), and 0.36 h (275 °C) (**b**), $N_2$ flow rate in 2,3-BDO dehydrogenation at a $W/F$ = 1.1 ($N_2$ flow at 30 cm$^3$ min$^{-1}$) and 1.5 h ($N_2$ flow at 60 and 90 cm$^3$ min$^{-1}$) (**c**), and reaction temperature in AC dehydrogenation at a $W/F$ = 2.3 (170 °C), 1.1 (200 °C), 0.73 (250 °C), and 0.73 h (275 °C) (**d**) over 12C4-2Cu/SiO$_2$ catalyst.

According to the $K_{1p}$ equation, reducing the partial pressure of the reaction system at a certain reaction temperature favors the production of AC, as the $K_{1p}$ value should remain constant [68]. This assumption is corroborated by the finding shown in Figure 3c. The increment of the $N_2$ flow rate reduced the partial pressure of the reaction system; thus, the equilibrium yield shifted to the product, which increased the yield of AC. Furthermore, the AC yield from the experimental results in Figure 3c, as indicated by the dot points, was in good agreement with the calculated values (red and blue broken lines). This finding indicates that the generation of Cu/SiO$_2$ catalysts by the use of organic additives gave outstanding catalytic activity for 2,3-BDO dehydrogenation, and the activity was only limited by the equilibrium. The AC yield and AC selectivity at 200 °C with an $N_2$ flow of 90 cm$^3$ min$^{-1}$ were 86.7 and 95.4%, respectively.

Figure 3d shows the results of AC dehydrogenation at different temperatures. The AC conversion was much lower than the 2,3-BDO conversion under the same reaction conditions, suggesting that AC dehydrogenation was relatively more difficult than 2,3-BDO dehydrogenation. The AC conversion and DA selectivity increased with the increase in reaction temperature. It is worth noting that 2,3-BDO was generated in the dehydrogenation of AC, and the 2,3-BDO selectivity decreased with the increase in reaction temperature. This result suggests that the hydrogen generated from the dehydrogenation of AC to DA was consumed for AC hydrogenation to produce 2,3-BDO at low reaction temperatures. In addition, the hydrogenation of AC was favorable at lower temperatures, while the AC dehydrogenation to generate DA proceeded dominantly at high reaction temperatures. Furthermore, the pressure equilibrium constant, $K_{2p}$, in the dehydrogenation of AC to DA was estimated to be 0.00025, 0.0024, 0.027, and 0.044 at 170, 200, 250, and 275 °C, respectively, assuming the reaction reaches an equilibrium where the $K_{2p}$ of the AC dehydrogenation is expressed as:

$$K_{2p} = \frac{P_{DA} \times P_{H_2}}{P_{AC}}$$

The findings in Figure 3d were in good agreement with the results in Figure 3b. Selective dehydrogenation of 2,3-BDO to AC proceeded at low reaction temperatures because the consecutive dehydrogenation of AC to DA hardly proceeded. On the other hand, the hydrogenation of AC to 2,3-BDO was inhibited since the substrate, 2,3-BDO, was abundant. Meanwhile, the AC generated from the first step of 2,3-BDO dehydrogenation was further dehydrogenated to produce DA at high reaction temperatures. Concurrently, the AC hydrogenation did not proceed since this reaction was favorable at low reaction temperatures.

### 3.4. Stability of Cu/SiO$_2$ Catalysts

Figure 4a compares the time course in the catalytic performance of none- and 12C4-10Cu/SiO$_2$ at 200 °C and a $W/F$ of 0.14 h. Both none- and 12C4-10Cu/SiO$_2$ catalysts deactivated since the 2,3-BDO conversion over both catalysts decreased. The none-10Cu/SiO$_2$ catalyst dropped the catalytic performance from 55 to 10% at a TOS of 5 h. Meanwhile, the 12C4-10Cu/SiO$_2$ catalyst dropped its catalytic performance from 78 to 39% at a TOS of 12 h. The results indicate that the 12C4-10Cu/SiO$_2$ catalyst had better performance than the none-10Cu/SiO$_2$ catalyst.

Figure 4b shows the XRD profiles of the spent none- and 12C4-10Cu/SiO$_2$ catalysts. According to the XRD profiles of the spent none-10Cu/SiO$_2$ catalyst, the Cu particle agglomeration did not proceed, as the crystallite size of the Cu metal before and after the reaction is nearly similar. On the other hand, the Cu nanoparticles in the 12C4-10Cu/SiO$_2$ catalyst aggregated as the Cu metal in the spent catalyst and were possibly larger than that freshly reduced 12C4-10Cu/SiO$_2$ catalyst, for which the XRD peak was too broad for the calculation of the crystallite size. Nevertheless, the crystallite size of the Cu metal in the spent 12C4-10Cu/SiO$_2$ catalyst after a TOS of 12 h was significantly smaller than even the freshly reduced none-10Cu/SiO$_2$ catalyst. These results indicate that the utilization of 12C4 as an organic additive is beneficial for catalytic properties, as it generates a Cu/SiO$_2$ catalyst with Cu nanoparticles even after the reaction proceeds.

Figure 4c depicts the TG curves of the spent none- and 12C4-10Cu/SiO$_2$ catalysts. The weight changes of the none- and 12C4-10Cu/SiO$_2$ catalysts were similar. This result shows that the amount of coke deposited on both catalysts was the same (8.9 wt.-%). It was reasonable to expect that both none- and 12C4-10Cu/SiO$_2$ catalysts might have similar catalyst deactivation rates. However, due to its high intrinsic activity, it spent a long time for the 12C4-10Cu/SiO$_2$ catalyst to drop its conversion to the same level as the none-10Cu/SiO$_2$ catalyst. In general, the utilization of organic additives in the catalyst preparation generated Cu/SiO$_2$ catalysts with better performance due to a higher intrinsic activity, i.e., formation rate, than the Cu/SiO$_2$ catalyst prepared without organic additive.

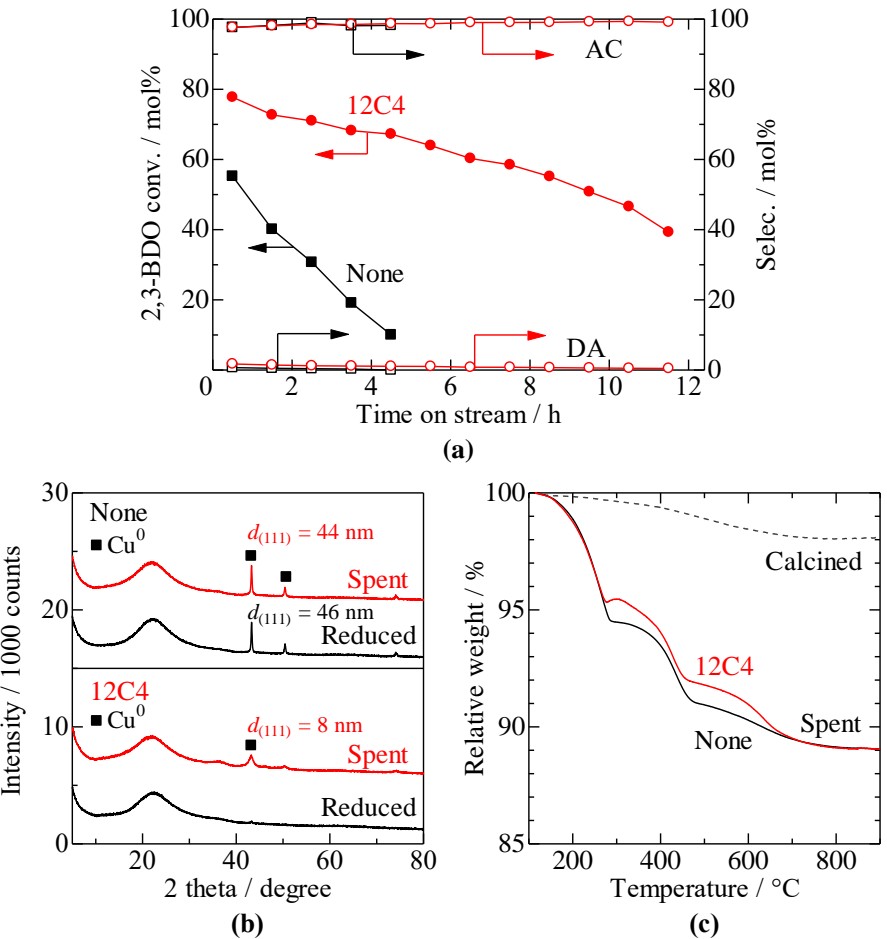

**Figure 4.** Time course of the catalytic performance in the dehydrogenation of 2,3-BDO at 200 °C and a $W/F$ of 0.14 h over none- and 12C4-10Cu/SiO$_2$ catalysts (**a**), XRD patterns of reduced and spent 10Cu/SiO$_2$ catalysts (**b**), and TG profiles of the spent none-, spent 12C4-, and calcined none-10Cu/SiO$_2$ catalysts (**c**).

*3.5. Effect of Cu Content and $SA_{Cu}$ on the Formation Rate of Acetoin*

Previously, we have reported the proportional relation between the formation rates of decanal and $SA_{Cu}$ of Cu/SiO$_2$ catalysts in the dehydrogenation of 1-decanol [59]. Figure 5a shows the effect of Cu content on the formation rate of AC per gram of Cu/SiO$_2$ catalyst, estimated at the 2,3-BDO conversion below 15% at low $W/F$. Increasing the Cu content of the 12C4-$x$Cu/SiO$_2$ catalysts increased the AC formation rate due to the increment of $SA_{Cu}$, which is summarized in Table S2. Similarly, the increase in Cu content of the none-$x$Cu/SiO$_2$ catalyst from 2 to 15 wt.-% improved the formation rate of AC. However, further increment of Cu content from 15 to 20 wt.-% decreased the formation rate of AC as the Cu agglomerated, leading to the formation of AC over the Cu/SiO$_2$ catalyst with smaller $SA_{Cu}$.

Figure 5b depicts the relation between $SA_{Cu}$ and the formation rate of AC in the dehydrogenation of 2,3-BDO. The formation rate of AC is proportional to the $SA_{Cu}$. However, the formation rate over 12C4-$x$Cu/SiO$_2$ catalysts was substantially higher than that over none-$x$Cu/SiO$_2$ catalysts due to a significantly higher $SA_{Cu}$. The linear relation between $SA_{Cu}$ and the catalytic activity of Cu catalysts has also been reported in CO$_2$ hydrogenation [69] and diols dehydrogenation [70], indicating that the reaction is only catalyzed by the surface of Cu metal. A similar proportional relation between $SA_{Cu}$ and the formation rate of GBL in the dehydrogenation of 1,4-BDO was also found, as described later.

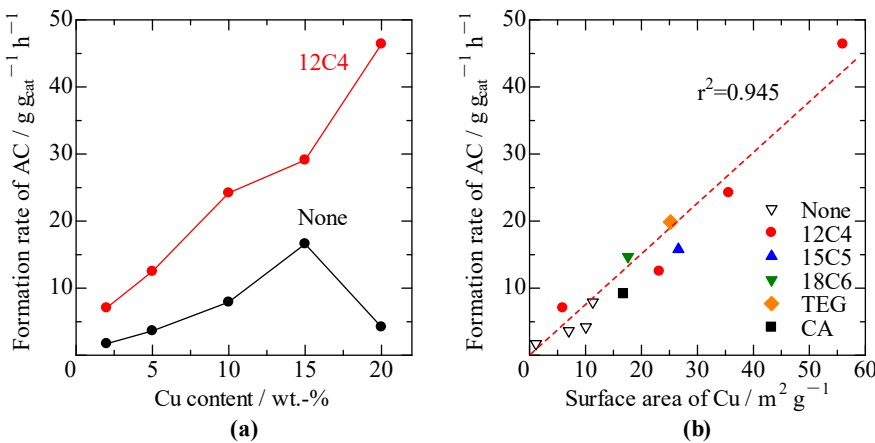

**Figure 5.** Effect of Cu content (**a**) and $SA_{Cu}$ (**b**) on formation rate of AC at 200 °C.

### 3.6. Dehydrogenation of 1,4-BDO over Cu/SiO$_2$ Catalyst

The high intrinsic activity of the 12C4-$x$Cu/SiO$_2$ catalysts in the dehydrogenation of 2,3-BDO was limited by equilibrium. Even though increasing the N$_2$ flow rate increased the equilibrium yield of AC, the equilibrium inhibited the full conversion of 2,3-BDO. The dehydrogenation of 1,4-BDO to GBL was performed to investigate the activity of the 12C4-$x$Cu/SiO$_2$ catalyst for 1,4-BDO dehydrogenation, which is an irreversible reaction (Scheme 1b) [17,44,45]. Figure 6a shows the time course of the catalytic performance of none- and 12C4-10Cu/SiO$_2$ catalysts in the dehydrogenation of 1,4-BDO at 260 °C and a $W/F$ of 0.080 h. Both catalysts maintained their 1,4-BDO conversion and GBL selectivity for 10 h; nevertheless, 12C4-10Cu/SiO$_2$ catalyst gave significantly higher catalytic activity than none-10Cu/SiO$_2$ catalyst. Figure 6b depicts the TG curves of the spent none- and 12C4-10Cu/SiO$_2$ catalysts. The coke formation occurred on both 12C4- and none-10Cu/SiO$_2$ catalysts, and the amounts of coke deposited on the spent 12C4- and none-10Cu/SiO$_2$ catalysts were 6.1 and 5.6 wt.-%, respectively. These findings indicate that the coke formation during 1,4-BDO dehydrogenation did not deactivate both 12C4- and none-10Cu/SiO$_2$ catalysts.

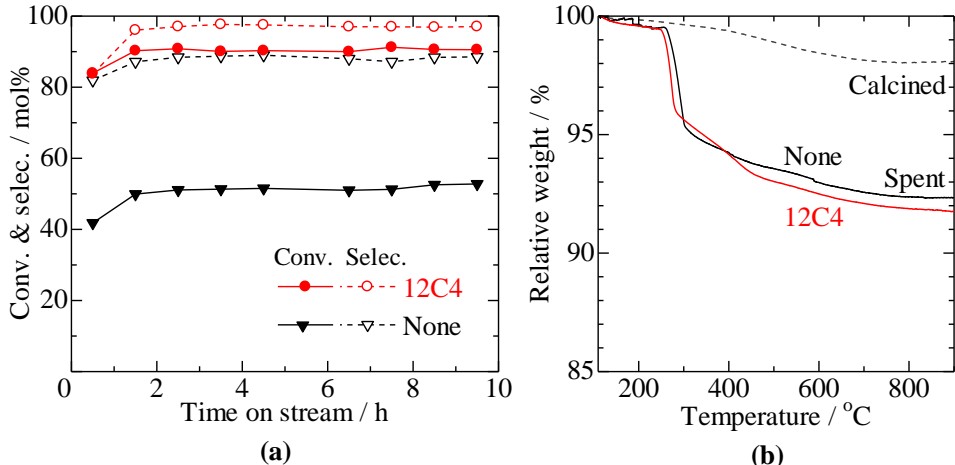

**Figure 6.** Catalytic performance at 260 °C and a $W/F$ of 0.080 h over none- and 12C4-10Cu/SiO$_2$ (**a**) and TG profiles of the spent none-, spent 12C4-, and calcined none-10Cu/SiO$_2$ catalysts (**b**).

Figure 7a depicts the activity of the none- and 12C4-10Cu/SiO$_2$ catalysts in the dehydrogenation of 1,4-BDO at different $W/F$. A high 1,4-BDO conversion and GBL selectivity were achieved by the use of the 12C4-10Cu/SiO$_2$ catalyst at a $W/F$ of 0.27 h. Under similar reaction conditions, the none-10Cu/SiO$_2$ catalyst had lower activity than the 12C4-10Cu/SiO$_2$ catalyst, showcasing the benefit of generating Cu/SiO$_2$ catalyst with highly

dispersed Cu nanoparticles using 12C4 as an organic additive. Figure 7b reveals that the dehydrogenation of 1,4-BDO was affected by the Cu content, with the 12C4-15Cu/SiO$_2$ catalyst giving the highest conversion among the Cu contents tested.

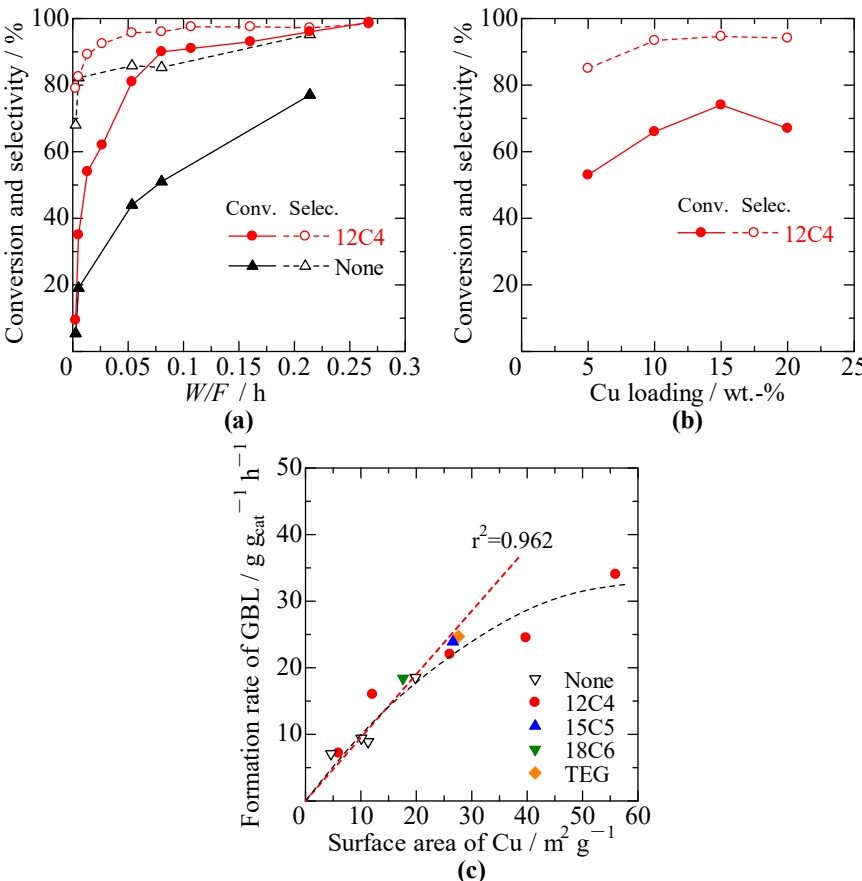

**Figure 7.** Effect of contact time, *W/F,* in the dehydrogenation of 1,4-BDO to GBL over 10Cu/SiO$_2$ catalyst at 260 °C (**a**), effect of Cu content at 240 °C and *W/F* = 0.054 h (**b**), and relation between Cu surface area on formation rate of GBL at 240 °C (**c**).

Figure 7c depicts the relation between $SA_{Cu}$ and the formation rate of GBL in the dehydrogenation of 1,4-BDO at 240 °C. The GBL formation rate is proportional to the $SA_{Cu}$ at $SA_{Cu}$ smaller than 30 m$^2$ g$^{-1}$, while the proportional relation was not observed at $SA_{Cu}$ higher than 30 m$^2$ g$^{-1}$. This phenomenon differed from the results in the dehydrogenation of 2,3-BDO to AC, in which the AC formation rate was proportional to $SA_{Cu}$, even at $SA_{Cu}$ higher than 30 m$^2$ g$^{-1}$. This difference can be explained by the mechanism of 1,4-BDO dehydrogenation to GBL. The dehydrogenation of 2,3-BDO to AC is a straightforward reaction, whereas 1,4-BDO dehydrogenation to GBL proceeds via a series of consecutive reactions, including (1) dehydrogenation of 1,4-BDO to 4-hydroxybutanal, (2) intramolecular hemiacetal-formed cyclization to 2-hydroxytetrahydrofuran, and (3) dehydrogenation of 2-hydroxytetrahydrofuran to form GBL (Scheme 2) [41]. The cyclization via an intramolecular hemiacetal reaction was possibly catalyzed by the acid sites of the silanol group in a similar manner to the cyclization of levulinic acid to angelica lactone [71]. Similarly, acidic alumina-supported Cu was also effective for the cyclization of 4-hydroxybutanal; nevertheless, the strong acidity of alumina promoted the dehydration reaction, generating tetrahydrofuran as the side product [41]. At $SA_{Cu}$ below 30 m$^2$ g$^{-1}$, the increment of Cu

content did not significantly alter the concentration of silanol sites; thus, the increment of Cu content favored the dehydrogenation of 1,4-BDO to 4-hydroxybutanal but did not hinder the consecutive cyclization of 4-hydroxybutanal to 2-hydroxytetrahydrofuran. However, when the $SA_{Cu}$ was higher than 30 m$^2$ g$^{-1}$, the increment of Cu content decreased the contribution of OH to the level that it slightly hindered the cyclization of 4-hydroxybutanal to 2-hydroxytetrahydrofuran and the subsequent GBL formation. As a result, a proportional relation between $SA_{Cu}$ and GBL formation rate was no longer observed at $SA_{Cu}$ above 30 m$^2$ g$^{-1}$, as shown in Figure 7c.

**Scheme 2.** Consecutive reaction from 1,4-BDO to GBL.

### 3.7. Vapor-Phase Oxidant-Free Dehydrogenation over Cu Catalysts

Table 2 shows the catalytic activity of several Cu catalysts in the dehydrogenation of 2,3-BDO to AC under various conditions. It is reported that several Cu/SiO$_2$ catalysts are active for the dehydrogenation of 2,3-BDO to AC [37,38]. The 12C4-10Cu/SiO$_2$ catalyst productivity was also higher than Cu catalyst-derived CuZnAl LDH [40], even when the reaction was performed at lower reaction temperatures. In the previous papers [38,40], it is notable that high AC selectivity is realized at a high temperature of 260 °C or higher if the formation of DA can be prevented at such high temperatures. The AC productivity of the 12C4-10Cu/SiO$_2$ catalyst was also higher than that of Cu-Al$_2$O$_3$ [39] at the same reaction temperature of 170 °C.

**Table 2.** Comparison of the productivity of AC over various reported Cu catalysts.

| Catalyst | Temp. /°C | *W/F* /h | TOS /h | Conv. /mol% | AC Selec. /mol% | AC Prod. /g g$^{-1}$ h$^{-1}$ | Ref. |
|---|---|---|---|---|---|---|---|
| 20.3Cu/SBA-15 | 280 | - | - | 9.9 * | >90 | - | [37] |
| 20Cu/SiO$_2$-AE [b] | 280 | - | 1 | 76.0 | 94.5 | - | [38] |
| Li-CuZnAl | 260 | 10.0 | 100 | 72.4 | 95.6 | 0.07 | [40] |
| 15Cu-Al$_2$O$_3$ | 170 | 4.00 | - | 63.0 | 96.0 | 0.15 | [39] |
| 12C4-10Cu/SiO$_2$ | 200 | 0.14 | 12 | 60.9 | 98.7 | 4.20 | This work |
| 12C4-2Cu/SiO$_2$ | 200 | 0.71 | 5 | 76.9 | 97.6 | 1.04 | This work |
|  | 170 | 1.43 | 5 | 53.9 | 99.2 | 0.36 |  |

* Reaction rate of 2,3-BDO in mmol min$^{-1}$ g$^{-1}$. [b] AE: prepared by ammonia evaporation.

Table 3 shows the comparison of the catalytic activities of several Cu catalysts in the vapor-phase dehydrogenation of 1,4-BDO to GBL. The utilization of 12C4 as an organic additive was proven to generate a Cu catalyst with the largest $SA_{Cu}$. Several Cu catalysts are highly active for the dehydrogenation of 1,4-BDO to GBL at 240 or 250 °C and a *W/F* of 0.5 h or higher [17,18,41,43–45]; nevertheless, the 12C4-10Cu/SiO$_2$ catalyst in our work gave a comparable catalytic performance at a shorter *W/F* of 0.27 h at 260 °C. It is also worth noting that the GBL productivity of 12C4-10Cu/SiO$_2$ was significantly higher than the GBL productivity of the reported Cu catalysts. In summary, catalytic activity in the dehydrogenation was improved by preparing a Cu/SiO$_2$ catalyst using organic additives. The substantial increment of $SA_{Cu}$ strongly influenced the increment of the catalytic activity.

**Table 3.** Comparison of the productivity of GBL over various reported Cu catalysts.

| Catalyst | $SA_{Cu}$ /m$^2$ g$^{-1}$ | Temp. /°C | W/F /h | TOS [a] /h | Conv. /mol% | Select. /mol% | GBL Prod. /g g$_{cat}^{-1}$ h$^{-1}$ | Ref. |
|---|---|---|---|---|---|---|---|---|
| Cu/ZnO/ZrO$_2$/Al$_2$O$_3$ [b] | 40.3 | 240 | 0.08 | 5 | 84 | 98 | 9.63 | [41] |
| 12Cu/SiO$_2$ | 3.90 | 250 | 0.50 | 10 | 100 | 98 | 1.92 | [17] |
| 10Cu/La$_2$O$_3$/ZrO$_2$ | 8.40 | 250 | 1.00 | 13 | 97 | 96 | 0.91 | [43] |
| 10Cu/SBA-15 | 8.68 | 250 | 1.00 | 17 | 100 | 98 | 0.95 | [44] |
| 10Cu/CeO$_2$ | 5.60 | 240 | 0.50 | 1 | 93 | 98 | 1.78 | [18] |
| 10Cu/CeO$_2$-Al$_2$O$_3$ | - | 240 | 0.50 | 1 | 100 | 99 | 1.94 | [45] |
| 12C4-10Cu/SiO$_2$ | 26.1 | 260 | 0.27 | 5 | 98.8 | 98.5 | 3.44 | This work |
|  |  |  | 0.054 | 5 | 81.4 | 95.7 | 13.8 |  |

[a] Time on stream in a flow system. [b] Prepared by co-precipitation; Cu content, 41.8 wt.-%.

## 4. Conclusions

Cu/SiO$_2$ catalysts prepared via the crown-ether-assisted impregnation method showed high catalytic activity in the vapor-phase oxidant-free dehydrogenation of 2,3- and 1,4-BDO. A 12C4 generated Cu/SiO$_2$ catalyst with the highest catalytic activity among several organic additives was tested. It was found that the increment of catalytic activity was strongly correlated with the generation of highly dispersed Cu nanoparticles. Since the formation rates of AC and GBL are proportional to $SA_{Cu}$, it is reasonable to assume that the improvement in $SA_{Cu}$ due to the utilization of 12C4 enhances the catalytic activity. The equilibrium nature of 2,3-BDO dehydrogenation inhibited the full conversion of 2,3-BDO to AC. Meanwhile, a nearly quantitative yield of GBL with high productivity was achieved in the irreversible dehydrogenation of 1,4-BDO by the use of 12C4-$x$Cu/SiO$_2$ catalysts.

**Supplementary Materials:** The following supporting information can be downloaded at: https://www.mdpi.com/article/10.3390/chemistry5010030/s1. Experimental details on the estimation of copper surface area Table S1: Detail quantity for the preparation of Cu/SiO$_2$ catalysts; Table S2: $SA_{Cu}$ values of Cu/SiO$_2$ catalysts used in Figures 5 and 7. References [59,61,72–75] are cited in Supplementary Materials.

**Author Contributions:** Conceptualization, S.S.; methodology, S.S.; validation, Y.Y. and S.S.; formal analysis, S.H. and M.K.; investigation, S.H. and M.K.; resources, S.S.; data curation, E.K., S.H. and M.K.; writing—original draft preparation, E.K.; writing—review and editing, Y.Y. and S.S.; visualization, E.K. and S.H.; supervision, Y.Y.; project administration, S.S.; funding acquisition, S.S. All authors have read and agreed to the published version of the manuscript.

**Funding:** This research was funded by the Japan Society for the Promotion of Science (JSPS KAK-ENHI, Grant Number JP21H01711).

**Institutional Review Board Statement:** Not applicable.

**Informed Consent Statement:** Not applicable.

**Data Availability Statement:** The data is contained within the article or supplementary material.

**Conflicts of Interest:** The authors declare no conflict of interest.

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
