# Peer review of "Vapor-Phase Oxidant-Free Dehydrogenation of 2,3- and 1,4-Butanediol over Cu/SiO2 Catalyst Prepared by Crown-Ether-Assisted Impregnation"

_chemistry, doi:10.3390/chemistry5010030_

Round 1

Reviewer 1 Report

This manuscript describes very nice piece of research that is obviously adequate for this journal. I found it very well organized, including an Experimental Section with a high degree of detail, although not without flaws as I will address later. The Discussion is well conducted with a critical point of view and the Conclusions are set adequately. Despite this, my questions are as follows:

1. In the Experimental Section, the preparation of the catalysts lacks details as authors did not report specific quantities and concentrations of the solution that was used to dope the SiO2 support, allowing others to easily replicate the synthesis. This must be done.

2. In Figure 1, authors show the TD-MS profiles, which for the 12C4-2Cu/SiO2 catalyst show a series of ions being followed. The outcomes are obvious with the treated catalyst displaying more ions than the none- counterpart. However, authors mention that "signals at m/z = 43 and 44 in the as-prepared 12C4-2Cu/SiO2 would be attributed from CH2CHO and CH2CH2O, respectively, whereas the signal at m/z = 44 in the as-prepared none-2Cu/SiO2 might originate from CO2." So, if for the latter the m/z=44 is due to CO2, then in the former the signal due to that m/z value should be due to both CH2CH2O and CO2. this needs to be corrected.

3. In Figure 1c, authors show the TG of both none- and 12C4-2Cu/SiO2 catalysts. However, there is a third profile that is not mentioned in the caption. What is it due and the caption must be updated.

4. In Section 3.2., authors must define "contact time, W/F" and what it represents, as it is different from time on stream.

5. Data in Figure 3 are defying to interpret, especially because one cannot understand the caption. Figure 3b I could not understand for what W/F the data refers to (0.36?) And what are the broken lines referring to? In Figure 3c, no information is provided on which curve refers to which W/F setpoint. The caption for Figure 3d mentions several W/F values and temperature, but one cannot understand for which W/F the plots refer to. Please revise both the Figure and caption.

6. In Figure 4c, authors show the profiles of both spent none and 12C4 catalysts. There is a third plot named "Fresh 12C4", which I do not understand what it is. If it is the real fresh catalyst, then this profile is completely different from that in Figure 1c. What is the reason? 

7. Figure 6b has the same problem as I mentioned just above for Figure 4c.

8. Figure 7a mentions in its legend some broken lines as being due to Selectivity plots. However, the Figure does not show any selectivity plots.  A similar situation is observed for Figure 7b, which mentions Conversion plots but none is there. Please revise both figures.

9. There is inconsistency across the 12C4 catalyst throughout the manuscript. Up to Section 3.3. authors mention 12C4-2Cu/SiO2. Then from Section 3.4. onwards it changes to 12C4-10Cu/SiO2 and sometimes simply 12C4-Cu/SiO2 (line 267). This makes reading and keeping track of thins really difficult. Please revise mandatorily.

Therefore, major revision is required.

Author Response

I attached a response sheet for Reviewer 1. 

Reviewer 2 Report

This work describes the use of heterogeneous catalysts for the dehydrogenation of 2,3 and 1,4-butanediols. From the formal point of view, the work is almost flawless. The catalysts are properly characterized, and the same can be said for the study of the reaction progress. The results are logically organized and discussed. Therefore, I have a minimum of comments:

Some papers describing the dehydrogenation of butanediols are missing in the introductory section:

https://doi.org/10.1021/om048983m
DOI https://doi.org/10.1039/B109658N
https://doi.org/10.1023/B:REAC.0000034836.56895.a9
https://doi.org/10.1016/j.ijhydene.2020.03.021
https://doi.org/10.1016/j.catcom.2008.12.014
https://doi.org/10.1016/j.molcata.2007.03.034

Minor typos:
line 98: there is "1h" should be "1 h"
Figure 6: there is "Temrperature..." Should be "Temperature"

Author Response

I attached a response sheet for Reviewer 2.

Round 2

Reviewer 1 Report

Authors have revised their manuscript and it can now be accepted for publication.